# PhotoGate microscopy to track single molecules in crowded environments

Vladislav Belyy[1,*], Sheng-Min Shih[2,*], Jigar Bandaria[2], Yongjian Huang[1,3], Rosalie E. Lawrence[3], Roberto Zoncu[3] & Ahmet Yildiz[1,2,3]

Tracking single molecules inside cells reveals the dynamics of biological processes, including receptor trafficking, signalling and cargo transport. However, individual molecules often cannot be resolved inside cells due to their high density. Here we develop the PhotoGate technique that controls the number of fluorescent particles in a region of interest by repeatedly photobleaching its boundary. PhotoGate bypasses the requirement of photo-activation to track single particles at surface densities two orders of magnitude greater than the single-molecule detection limit. Using this method, we observe ligand-induced dimerization of a receptor tyrosine kinase at the cell surface and directly measure binding and dissociation of signalling molecules from early endosomes in a dense cytoplasm with single-molecule resolution. We additionally develop a numerical simulation suite for rapid quantitative optimization of Photogate experimental conditions. PhotoGate yields longer tracking times and more accurate measurements of complex stoichiometry than existing single-molecule imaging methods.

[1] Biophysics Graduate Group, University of California, Berkeley, California 94720, USA. [2] Department of Physics, University of California, Berkeley, California 94720, USA. [3] Department of Molecular Cell Biology, University of California, Berkeley, California 94720, USA. * These authors contributed equally to this work. Correspondence and requests for materials should be addressed to A.Y. (email: yildiz@berkeley.edu).

Single-particle tracking (SPT) avoids ensemble averaging and allows the stochasticity and heterogeneity of molecular behaviour to be observed directly in real time. Performing these experiments in live cells has been a challenging task because of the crowded nature of cellular environments, the short time scales of molecular dynamics and the lack of appropriate microscopy tools. In conventional microscopy, molecules closer to each other than the diffraction limit ($\sim 250$ nm for visible light)[1] cannot be readily resolved due to the overlap of their point spread functions. Therefore, the concentration limit for single-molecule detection is $\sim 1$ nM, which corresponds to a few molecules in an entire *Escherichia coli* cell. Molecules separated by less than the diffraction limit can be resolved using stochastic photoactivation[2,3], structured illumination[4] and stimulated emission depletion[5]. However, breaking the diffraction limit comes at the expense of lower temporal resolution either through image scanning[5] or super-positioning of multiple frames[2–4], which make these techniques unsuitable for monitoring rapid dynamic processes.

Photobleaching and photoactivation of fluorescent probes are often used to study the dynamics of molecules inside cell. Single-particle tracking photoactivation light microscopy (sptPALM) overcomes the concentration limit for SPT by photoconverting a small subset of 'dark' molecules at a time[6]. While this method achieves high frame rates, it only yields very short trajectories (typically $<20$ frames)[7] because of the low photon budget of photoconvertible probes[8]. Photobleaching is often used to reduce the apparent concentration of fluorophores[9] and to observe the dynamics of molecules inside cells by monitoring fluorescence recovery after photobleaching (FRAP)[10]. Because the fluorescent particles arrive in a continuous stream from the unbleached region in FRAP assays, single particles can be resolved only at the very onset of the recovery process, using thinning out clusters while conserving stoichiometry of labelling (TOCCSL). Effective SPT time of TOCCSL is limited by fluorophore density and the fluorescence recovery rate[11]. This rapid recovery makes FRAP-based approaches ill-suited for observing dynamic interactions between individual proteins and large cellular structures, such as organelles and cytoskeleton.

In this study, we develop PhotoGate to track single particles in dense specimens for extended periods of time. The method selectively photobleaches fluorescent particles and controls the number of fluorophores that enter the region of interest (ROI). Unlike FRAP, new fluorescent particles streaming into the ROI are repeatedly photobleached at its boundary by a gate beam and the density of fluorophores in the ROI remains at a constant low level. Therefore, PhotoGate bypasses the need for photoactivation and substoichiometric labelling, and achieves SPT of conventional probes until photobleaching at concentrations two orders of magnitude higher than the diffraction limit. Using this method, we detect arrival and departure events of single APPL1 molecules on the surface of early endosomes, and the monomer-to-dimer transition of epidermal growth factor receptor (EGFR) receptors in response to a signalling cue on a mammalian cell membrane.

## Results

**The PhotoGate assay.** PhotoGate utilizes two separate laser beams of the same excitation wavelength and orthogonal polarization (Supplementary Fig. 1a). The first laser beam, referred to as the gate beam, is focused in the image plane and swept outwards from the centre in a spiral pattern to pre-bleach a circular ROI (Fig. 1a; Supplementary Movie 1). The gate beam is then shuttered for several seconds to allow a small number of unbleached molecules to diffuse into the ROI. The second beam, referred to as an imaging beam, is used to image fluorescent

molecules in the ROI at a high signal-to-noise ratio (SNR) under total internal reflection (TIR) illumination. To prevent additional fluorescent molecules from diffusing into the ROI during imaging, the gate beam is repeatedly swept along the outer perimeter of the ROI. As a result, the density of fluorescent objects in the ROI remains low and does not increase over time.

We found that sweeping a 15-µm diameter area with a focused gate beam (10 MW cm$^{-2}$, 1 µm full-width half-maximum) in 10 s was sufficient to photobleach nearly all of the fluorescent molecules in the ROI (Fig. 1b; Supplementary Fig. 2). During pre-bleaching, the background fluorescence in the ROI was greatly reduced without significantly ($<20\%$) photobleaching the molecules outside the ROI (Fig. 1c,d). The diameter of the imaging beam matches that of the pre-bleached ROI (Supplementary Fig. 1; Supplementary Movie 2) to prevent useless bleaching of the fluorescent molecules outside the ROI.

**Measuring APPL1 residence times on early endosomes.** PhotoGate is uniquely suited for observing the dynamic interactions of concentrated and rapidly diffusing molecules with relatively immobile structures, such as large organelles in a cytoplasm. To demonstrate this class of applications, we used PhotoGate to measure the single-molecule residence times of APPL1, a mediator of intracellular EGFR signalling, on the surface of endosomes. APPL1 transiently localizes to a subset of vesicles created by clathrin-mediated endocytosis or micropinocytosis, and marks early endosomes before their conversion to the phosphatidylinositol 3-phosphate (PI3P) stage[12] (Fig. 2a). It is unclear whether APPL1 binds tightly to its target vesicles and only dissociates after receiving a specific signalling clue, or the bound and unbound populations remain in dynamic equilibrium.

To detect the arrival and departure events of individual APPL1 molecules at early endosomes, we transiently transfected human U2OS osteosarcoma cells with a C-terminal green fluorescent protein (GFP) fusion of APPL1 (APPL1–GFP). The cells appeared bright with background fluorescence, with endosomes visible as even brighter spots (Fig. 2b). Cells were pretreated with nocodazole to immobilize the endosomes (Methods) and to confirm that the observed single-molecule arrival and departure events are spatially correlated with the last known endosome positions (Supplementary Fig. 3). The high concentration of APPL1 both in the cytoplasm and on the surface of endocytic vesicles makes it impossible to observe its dynamics using conventional TIR fluorescent (TIRF) imaging. We first performed FRAP by fully bleaching a 17 µm diameter ROI and imaging the recovery of fluorescence until steady state was reached under TIRF illumination (Fig. 2b; Supplementary Movie 3). During recovery, density of APPL1 at individual endosomes almost immediately exceeded the single-molecule detection limit, making FRAP-based approaches unsuitable for SPT of APPL1 (Fig. 2c). The mean lifetime of recovery at endosomes was found to be $24.7 \pm 4.5$ s (s.e.m.; Fig. 2d). Interestingly, the lifetime of recovery in an endosome-free region was only slightly lower than at endosomes, $21.3 \pm 2.7$ s (s.e.m.; Fig. 2d), suggesting that the recovery is diffusion limited. FRAP yields little information about the reaction rates of APPL1 at endosomes under these conditions, because it cannot independently measure the diffusion and reaction components[13].

PhotoGate bypasses this issue by directly measuring the dissociation of single APPL1 molecules from endosomes and eliminating the signal of rapidly diffusing molecules in cytoplasm. We bleached a 15 µM diameter ROI in the same way as in the FRAP experiment, followed by periodic sweeping the gate beam around the perimeter of the ROI every 2 s. The gate beam bleached the majority of fluorescent APPL1 molecules entering

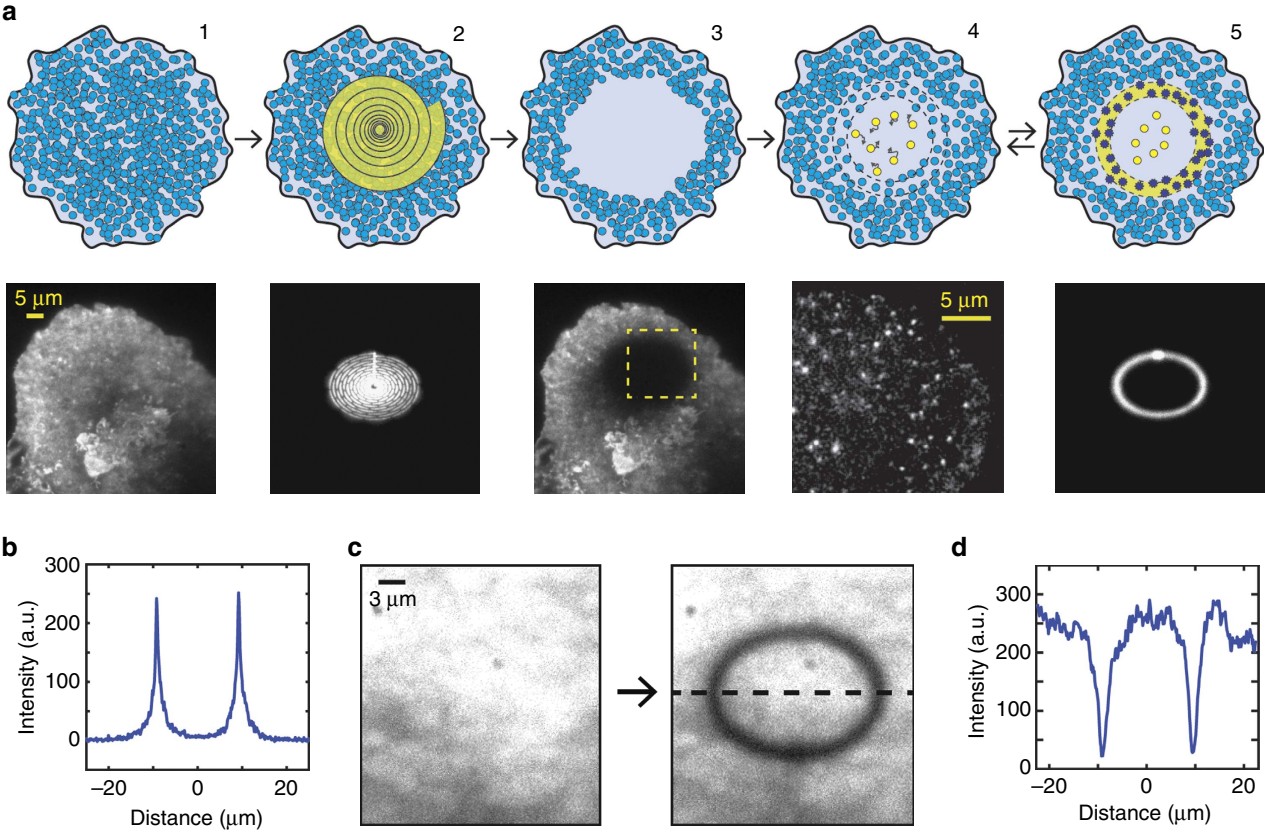

**Figure 1 | PhotoGate imaging for single-particle tracking** (**a**) 2D PhotoGate schematic (top row) with corresponding representative images (bottom row). (1) A cell surface is heavily decorated with fluorescently labelled molecules (blue circles). (2–3) The focused gate beam is swept outwards from the centre in a spiral pattern to pre-bleach an elliptical region. (4) The gate beam is turned off to allow diffusion of fluorescent molecules from the periphery of the ROI, which are then imaged under TIRF illumination (yellow circles). (5) The gate beam is repeatedly turned on to photobleach fluorescent particles entering the ROI (dark blue circles) while single molecules inside the ROI are imaged under TIRF illumination. Photobleached particles are not shown for clarity. (**b**) Linear intensity profile along a line bisecting the PhotoGate ring shown in **a**. (**c**) A coverslip densely coated with GFP is bleached with a single sweep of the PhotoGate ring to demonstrate the bleaching profile (**d**) The bleaching profile plotted along the black dashed line in **c**.

the ROI (Fig. 2e; Supplementary Movie 4), because fluorescence levels remained constantly low, while the gate was on and fully recovered within tens of seconds when the gate was turned off. The ROI was imaged under TIR illumination at two frames per second (a time-lapse imaging protocol consisting of 50 ms exposure followed by 450 ms dark time). We detected single fluorescent spots appear at and dissociate from the pre-bleached ROI (Fig. 2f). Ninety-five percent of reappearing spots ($N = 84$) localized within 300 nm of the centre of previously detected endosomes. Because the endosomes are sparsely distributed, the area covered by the 300 nm circles centred over each endosome occupies only $11 \pm 6\%$ of the entire cell's area. We concluded that reappearing spots represent binding of APPL1 to endosomes, rather than random binding to other locations in the cell ($P < 10^{-70}$). The residence time distribution of individual molecules (Fig. 2g) revealed that the mean residence time of APPL1 is 8.6 s (7.0–10.9 s, 95% conf. int.). Interestingly, this is several fold shorter than the FRAP recovery lifetime, further illustrating that FRAP data is often a convolution of multiple distinct processes that cannot be easily untangled.

To rule out the possibility that the observed disappearances of APPL1–GFP spots were a result of photobleaching rather than physical dissociation of APPL1 from endosomes, we repeated the experiment by keeping the exposure time constant at 50 ms and adjusting the dark time to alter the frame time to 150 and 1,500 ms. Had the observed 'departure' events been caused by

photobleaching, the measured residence times would depend primarily on the number of collected frames under the same TIRF excitation power. In contrast, we observed residence times to remain nearly independent of frame rate (Supplementary Fig. 4), demonstrating that they correspond to departures of APPL1–GFP from endosomes. These measurements suggest that the short-lived, dynamic association of single APPL1 proteins with the endosomal surface may enable the rapid and highly coordinated displacement of the entire APPL1 population as the endosome transitions to the PI3P-positive stage[12].

**Tracking EGFR diffusion on a mammalian cell membrane.** We next applied PhotoGate to reveal the dynamics of single receptor complexes on a live cell membrane. EGFR is a prototype of receptor tyrosine kinases[14], comprising an extracellular ligand-binding domain, a single-transmembrane helix and an intra-cellular tyrosine kinase domain (Fig. 3a). Binding of EGF to the extracellular domains promotes the formation of an asymmetric dimer of intracellular kinase domains, one of which allosterically activates the other[15] and triggers the recruitment of the down-stream signalling proteins. In fluorescence microscopy studies, ligand-induced dimerization was inferred indirectly from the differences in the diffusion constants between ligand-bound[16,17] and ligand-free[14,18] receptors. In these studies, labelling densities were kept several orders of magnitude lower than the actual

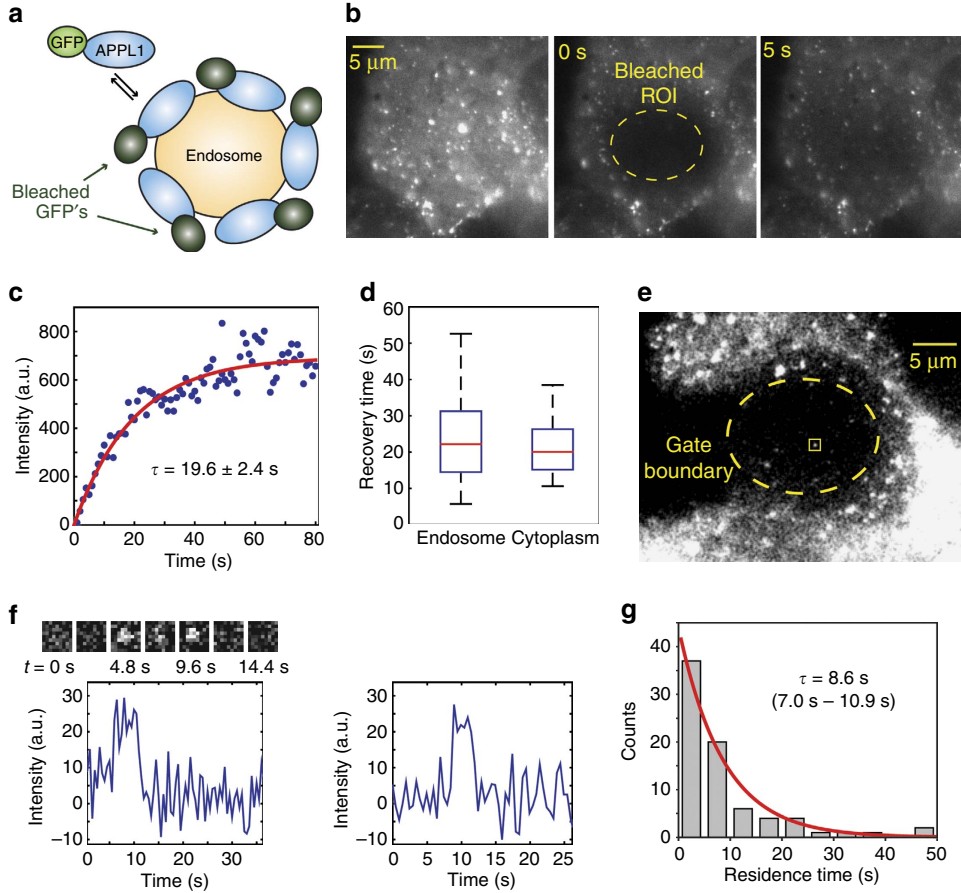

**Figure 2 | Measuring APPL1 residence times on early endosomes.** (**a**) A model for the exchange of fluorescent APPL1–GFP with bleached APPL1 molecules on an endosome. (**b**) (Left) image of a U2OS cell expressing APPL1–GFP under TIRF excitation. A 17 μm diameter area in the cytoplasm was photobleached at $t = 0$ s (middle), and recovery of GFP-APPL1 fluorescence within the bleached area was recorded using FRAP (right). (**c**) A typical FRAP curve of a single cell, measured at a single endosome after bleaching a 10 μm diameter region. The red curve represents a single exponential fit to the fluorescence recovery data (mean ± 95% confidence interval). (**d**) Recovery lifetimes, measured at endosomes and endosome-free regions of the cytoplasm. The line within the boxplot represents the median. The outer edges of the box are the 25th and 75th percentiles. The whiskers extend to the minimum and maximum values. $N = 10$ cells for each condition. (**e**) Image of an APPL1–GFP U2OS cell in the middle of a PhotoGate experiment. The yellow square represents the appearance of a single APPL1 molecule at a previously determined endosomal location in the ROI. (**f**) Fluorescence intensity trajectories of single spots inside the ROI reveal the arrivals and departures of single APPL1 molecules. (Top) enlarged snap shots at different time points of the fluorescent spot marked with a yellow square in **e**. (**g**) Residence times of single APPL1 molecules are fitted with a single exponential decay to obtain the average lifetime (mean and 95% confidence interval).

receptor densities on the cell membrane, which are too crowded for SPT. Therefore, majority of the receptor molecules remained unlabelled and the oligomeric state of EGFR molecules could not be directly determined by subunit counting.

We expressed the mNeonGreen-fusion of EGFR in monkey fibroblast (COS7) cells. Because the cell membrane was densely covered with mNeonGreen-EGFR (400 molecules per μm²), it was not possible to track individual molecules by conventional TIRF imaging (Fig. 3b), even during the onset of the fluorescence recovery process in FRAP assays (Fig. 3c; Supplementary Fig. 5; Supplementary Movies 5–7). Using FRAP, the recovery of the EGFR fluorescence into a 4 μm diameter ROI on a cell membrane was recorded at 100 ms temporal resolution (Fig. 3c; Supplementary Movie 5). The fluorescence recovery analysis revealed that the average diffusion constant of EGFR was $0.22 \pm 0.02\ \mu m^2 s^{-1}$ in serum-starved conditions (Fig. 3d).

Using PhotoGate, we simultaneously determined the diffusion constants and stoichiometry of individual EGFR complexes in real time. Initially, a 15-μm diameter ROI was pre-bleached by sweeping the focused gate beam, followed by a 2 s dark waiting period to allow a few fluorescent molecules to diffuse into the

ROI. After pre-bleaching, single mNeonGreen spots were tracked at a rate of 20 frames per second with an imaging beam (Fig. 3e; Supplementary Movies 2 and 8) without inducing harmful effects to living cells. Imaging was interspersed by 200-ms long sweeps of the gate beam around the outer perimeter of the ROI. On average, ~47,000 photons were detected from one mNeonGreen until photobleaching (Fig. 3f)[19]. The mean tracking lifetime ($\tau$) of single mNeonGreen-EGFR was $10.1 \pm 1.2$ s. During SPT, fluorescent molecules outside the ROI remain unbleached because they are not subjected to TIRF illumination.

The frequency of gating was adjusted depending on the diameter of the ROI, the width of the gate, and the density and diffusion constant of the fluorophores. The probability of particles escaping the gate was calculated based on the time dependent recovery profile in FRAP experiments[20] (Methods) and was kept low ($10^{-7}$) to prevent crowding of fluorophores inside the ROI.

To compare the tracking ability of PhotoGate to sptPALM, we replaced mNeonGreen with mEos2, one of the brightest photoconvertible fluorescent proteins[8]. We expressed mEos2-EGFR in COS7 cells under the same conditions and tracked

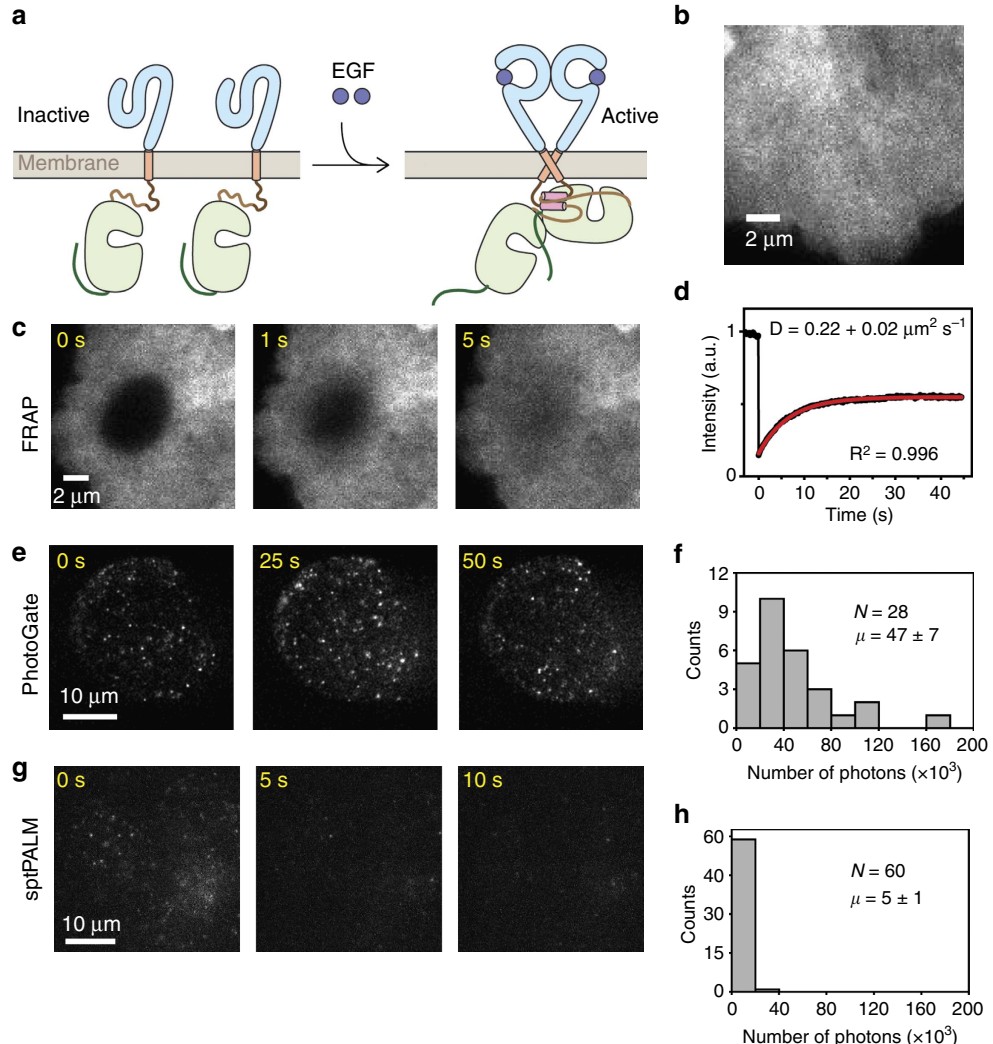

**Figure 3 | Tracking of single EGFR molecules in a live cell membrane** (**a**) A model for the ligand-induced dimerization of EGFR. EGFR monomer contains an extracellular ligand-binding region (blue), a single-transmembrane helix, an intracellular tyrosine kinase domain (light yellow) and a C-terminal tail. Binding of EGF ligands to the extracellular domain induces an asymmetric dimer formation. (**b**) mNeonGreen-EGFR molecules densely cover COS7 cell membrane. (**c**) mNeonGren-EGFR molecules exhibit a continuous recovery of fluorescent intensity in a FRAP assay. (**d**) The fit of fluorescence recovery signal (red curve) reveals the average diffusion constant of EGFR spots. (**e**) In PhotoGate, diffusion of single mNeonGreen-EGFR molecules was tracked in the ROI over 30 s. (**f**) On average, ~ 47,000 photons were detected from single mNeonGreen molecules using PhotoGate (mean ± s.e.m.). (**g**) In the sptPALM experiment, individual mEos2-EGFR molecules were photoactivated with 405 nm excitation at $t = 0$ s, and fluorescent spots were tracked over 5 s. (**h**) The number of photons detected from single mEos2 spots before photobleaching (mean ± s.e.m.) was nine times lower than that of mNeonGreen.

individual fluorescent spots using sptPALM at 20 frames per second (Fig. 3g; Supplementary Movie 9)[21]. On average, we detected ~ 5,000 photons from a single mEos2 spot before photobleaching ($\tau = 1.2$ s; Fig. 3h). The results showed that SPT of mNeonGreen spots using PhotoGate provides more than 10-fold higher photon budget and hence better tracking ability than sptPALM of mEos2 spots.

The diffusion constants of individual spots were estimated by linearly fitting their mean square displacement (MSD; Fig. 4a,b). Spots that display confined diffusion or unidirectional motion were excluded from the analysis (~ 7%; Supplementary Fig. 6a,b). In addition, 3–7% of EGFR complexes paused in the membrane for a few seconds (Supplementary Fig. 6c), which may reflect transient interactions between EGFR and membrane domains[14], as well as cortical actin meshwork[22,23].

The histogram of the diffusion constants of freely diffusing molecules revealed two distinct populations (Fig. 4c). In the absence of the EGF ligand in the serum, the majority (91%) of

EGFR stay in a more diffusive state ($D = 0.25\,\mu m^2\,s^{-1}$) and only 9% were found at a less diffusive state ($D = 0.12\,\mu m^2\,s^{-1}$, $N = 85$). We observed a large shift (84%, t-test, $P < 0.01$, $N > 80$) in population towards the less diffusive state on the addition of 16 nM EGF. The results show that the diffusion constant of EGFR molecules decreases nearly twofold on addition of EGF.

Photobleaching analysis of GFP-EGFR spots in COS7 cells revealed the stoichiometry of EGFR complexes (Fig. 4d). In the absence of EGF, 91% of the fluorescent spots ($N = 35$) displayed one-step photobleaching, indicating that they represent monomers of GFP-EGFR. In the presence of EGF, we observed that 57% of the spots ($N = 40$) bleach in two steps, suggesting that EGFR forms a dimer containing two GFPs per spot (Fig. 4e). The average intensities of spots that bleached in two steps were nearly double (2.1-fold higher without EGF and 1.8-fold higher with EGF) those of spots that bleached in a single step. In addition, spots that bleached in two steps diffused slower ($D = 0.13 \pm 0.04$ $\mu m^2\,s^{-1}$, s.d.) compared with spots with only one fluorophore

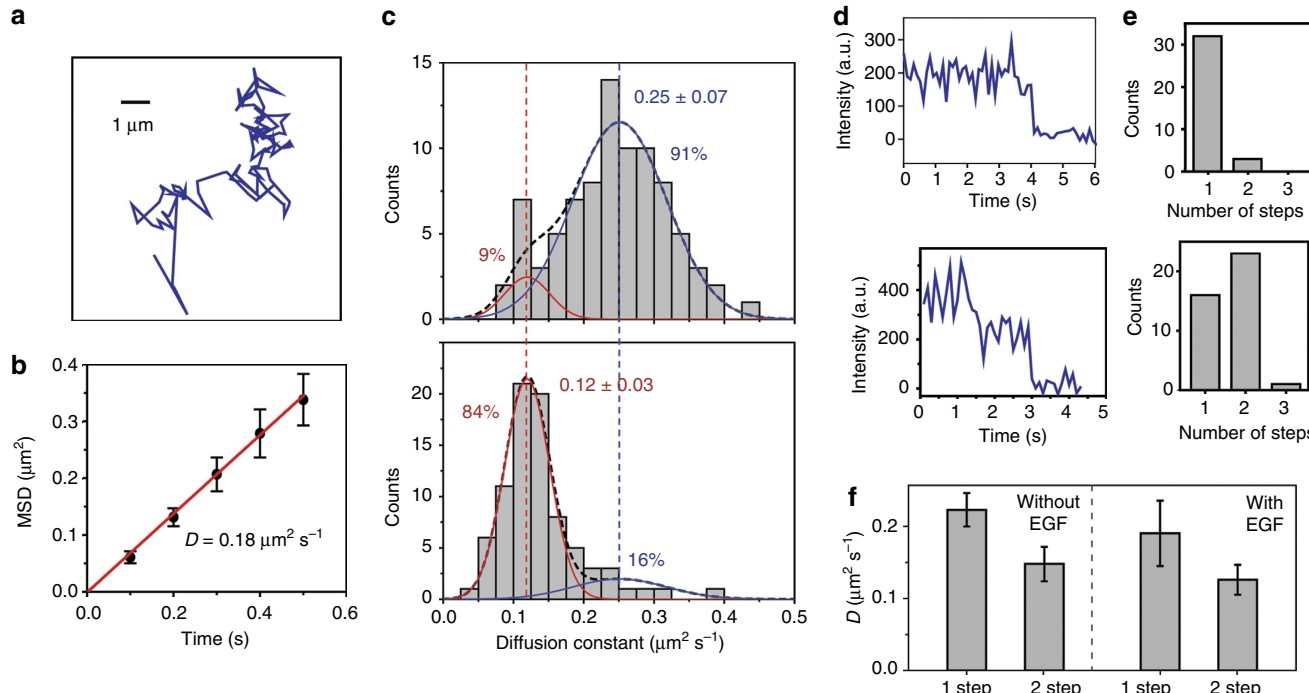

**Figure 4 | Ligand-induced dimerization of EGFR is determined by diffusion analysis and subunit counting.** (**a**) Sample trajectory of an individual GFP-EGFR spots diffusing on a COS7 cell membrane. (**b**) MSD plot of an example trajectory (mean ± s.d.). The slope of the linear fit (red line) represents the diffusion constant. (**c**) Diffusion constant histogram of EGFR spots in the absence (top) and the presence of 16 nM EGF. Multiple Gaussian fit (black dotted curve) reveals two major peaks. Without EGF, 91% of the spots are at the more diffusive state ($D = 0.25 \pm 0.07\,\mu m^2\,s^{-1}$, blue curve) and 9% at the less diffusive state ($D = 0.12 \pm 0.03\,\mu m^2\,s^{-1}$, red curve). EGF addition results in the 75% of the spots shifting from the more diffusive to the less diffusive state. (**d**) Example intensity profiles of EGFR spots showing one- (top) and two-step (bottom) photobleaching. (**e**) Photobleaching step histograms of GFP-EGFR spots in the absence (top) and presence (bottom) of EGF. (**f**) The average diffusion constant of EGFR spots that photobleached in one- and two-steps with and without EGF (mean ± s.e.m.).

($D = 0.21 \pm 0.07\,\mu m^2\,s^{-1}$; $t$-test, $P = 0.0036$; Fig. 4f), in agreement with the two-state distribution of the diffusion constant histograms (Fig. 4c). This result further indicates that the more diffusive particles are mostly monomers and the less diffusive particles are mostly dimers and higher-order oligomers[24]. We note that the percentage of molecules that show two-step bleaching (53%) is not as high as the percentage of molecules in the less diffusive state (84%) in 16 nM EGF. The single steps are partly due to premature bleaching as fluorescent spots pass near the gate; reversible dimerization and membrane dissociation of EGFR[25], and limited time resolution to detect multiple steps. Given that ~20% of GFP molecules remain in a dark state due to misfolding or incomplete maturation[26], we estimated that the probability of observing a two-step bleaching event for a dimer is ~50% under these conditions, consistent with our observation.

**Comparison of PhotoGate to TOCCSL by computer simulations.** The success of a PhotoGate experiment depends on a large number of parameters, some of which are controllable by the user (that is, ROI dimensions, laser intensity and gating frequency) and others inherent to the given biological system (that is, fluorophore density, diffusion constant and oligomerization state). We implemented a computer simulation of the PhotoGate experiment to optimize the imaging conditions for SPT. Densely crowded ($50\,\mu m^{-2}$) fluorescent particles emitting on average $5 \times 10^5$ photons before photobleaching were allowed to diffuse on a two-dimensional (2D) lattice representing the bottom surface of a cell. Photon flux of excited fluorophores was set to $2{,}000\,s^{-1}$. Photobleaching of individual spots was calculated for every frame

of the simulation with a Monte Carlo algorithm (Supplementary Tables 1–3; Methods). SPT trajectories were constructed for fluorescent particles that stayed at least $0.5\,\mu m$ away from any other unbleached fluorophore.

Using the parameters that represent our experimental conditions, simulations have revealed that PhotoGate is superior to TOCCSL in the number of usable trajectories that can be obtained from a single cell, the duration of trajectories and the ability to determine the oligomeric state of the molecules. For dimeric particles with a diffusion constant of $0.1\,\mu m^2\,s^{-1}$, the presence of a 0.5 Hz gate around the perimeter of the $14\,\mu m$ diameter ROI resulted in mean SPT tracking times of $0.83 \pm 0.02\,s$ (mean ± s.e.m.), four fold higher than $0.192 \pm 0.002\,s$ (mean ± s.e.m.) achievable by TOCCSL (Fig. 5a). Importantly, while the majority of trajectories are short in both experiments, the distribution of tracking times in the PhotoGate experiment includes a broad shoulder of particles that can be tracked for long periods of time. A number of $120 \pm 11$ particles per cell can be tracked for longer than 3 s with PhotoGate (mean ± s.d., $N = 10{,}100$ trajectories from five cells), compared with only $7 \pm 11$ in the equivalent TOCCSL experiment (mean ± s.d., $N = 23{,}468$ trajectories from five cells). This order of magnitude difference in long SPT trajectories stems from the fact that the ROI in the TOCCSL experiment gets crowded beyond the diffraction limit within seconds of experiment initiation. The difference in long-term tracking becomes even more pronounced when looking at more rapidly diffusing fluorophores with $D = 0.5\,\mu m^2\,s^{-1}$. Using a 3 Hz PhotoGate yields $25 \pm 17$ tracking trajectories longer than 5 s per cell (mean ± s.d., $N = 4{,}173$ trajectories from five cells), whereas such long trajectories were

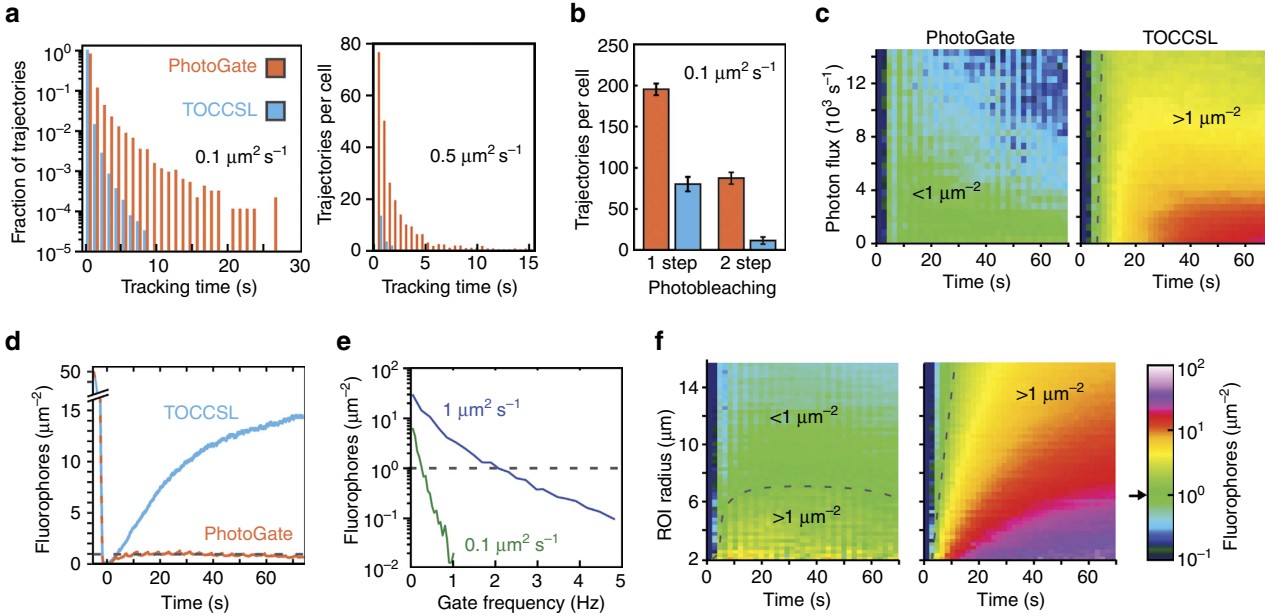

**Figure 5 | Numerical simulations of PhotoGate experiment and comparison with TOCCSL.** (**a**) Durations of single-particle trajectories with $D = 0.1\,\mu m^2\,s^{-1}$ with 0.5 Hz gating frequency (left) and particle trajectories with $D = 0.5\,\mu m^2\,s^{-1}$ with 3 Hz gating frequency (right) in a simulated PhotoGate experiment, compared to equivalent TOCCSL experiments. (**b**) Numbers of single particle trajectories of a dimeric molecule with $D = 0.1\,\mu m^2\,s^{-1}$ showing one- and two-step bleaching events over the course of an 80-second experiment in a single cell (mean ± s.d.). (**c**) Fluorophore density in a 14 μm diameter ROI over the course of a PhotoGate (left) and TOCCSL experiment (right) as a function of photon flux of a single fluorophore per second. (**d**) Density of fluorophores inside a 14 μm ROI in typical TOCCSL and PhotoGate experiments (0.5 Hz gating frequency and 1,500 photons per s). (**e**) Density of fluorophores inside the ROI of a PhotoGate experiment 30 s after initial bleaching as a function of gating frequency. Data are shown for fluorophores with diffusion constants of 1 and $0.1\,\mu m^2\,s^{-1}$. (**f**) Fluorophore density over the course of a PhotoGate (left) and TOCCSL (right) experiments as a function of ROI radius. Black arrow on the intensity scale and dotted lines in the heat maps point to approximate single-molecule detection limit (1 fluorescent spot $\mu m^{-2}$). In all simulations, the average number of photons detected per fluorophore before photobleaching is $5 \times 10^5$ photons, and surface density of fluorescent spots is $50\,\mu m^{-2}$ at the beginning of the simulations. See Supplementary Tables 1–3 for a detailed list of the simulation parameters.

not observed without a gate (0 out of 4,173 trajectories from five cells; Fig. 5a). Mean SPT tracking times remain higher in the PhotoGate experiment for the rapidly diffusing particles as well, at $0.746 \pm 0.024$ s compared with $0.145 \pm 0.002$ s (mean ± s.e.m.) without a gate. For the rest of comparative analysis of PhotoGate and TOCCSL, we focused on fluorophores diffusing at $0.1\,\mu m^2\,s^{-1}$, because TOCCSL does not produce sufficient SPT trajectories for molecules diffusing more rapidly than $0.5\,\mu m^2\,s^{-1}$. The longer tracking trajectories accessible to PhotoGate can substantially increase the accuracy and reliability of diffusion analysis, especially in conditions when the membrane is measurably heterogeneous and a single particle may experience local confinement events over the course of a single trajectory.

Next, the number of photobleaching steps were detected from unique tracking trajectories. We found that 31% of dimeric fluorophores exhibited two bleaching steps over a single uninterrupted tracking trajectory using PhotoGate, compared with the 12% in an equivalent TOCCSL simulation. Moreover, the total number of tracking trajectories bleaching in two steps was $87.3 \pm 7.1$ per cell in PhotoGate, while only $11.4 \pm 4.4$ trajectories were obtained using TOCCSL (Fig. 5b). Taken together, these data demonstrate that PhotoGate is significantly better suited for the determination of complex stoichiometry by photobleaching than TOCCSL, since more diffusing complexes can be tracked per cell and a larger fraction of the tracked complexes bleach in multiple steps.

Next, we tested how fluorophore density inside the ROI is affected by TIRF illumination intensity. We ran simulations by altering the flux of fluorophores from 0 to $12 \times 10^3$ photons per s and calculated the fluorophore density within ROI over the

course of 80 s. In a simulated PhotoGate experiment at 0.5 Hz gate frequency and 14 μm diameter ROI with dimeric fluorophores diffusing at $0.1\,\mu m^2\,s^{-1}$, fluorophore density quickly rose to $\sim 1\,\mu m^{-2}$, following the initial bleaching cycle and stayed at that level for over a minute (Fig. 5c; Supplementary Movie 10) nearly independent of photon flux. In the absence of a gate, nearly all TIRF illumination intensities resulted in a rapid flood of unbleached fluorophores into the ROI, quickly bypassing the density required for single-molecule detection (Fig. 5d; Supplementary Movie 11). Increasing the illumination intensity in TOCCSL slightly increases the mean SPT time, but also leads to rapid photobleaching of the molecules outside the ROI. This helps explain why longer tracking trajectories are achievable with PhotoGate (Fig. 5a). In the absence of a gate, density of fluorophores increases rapidly after bleaching such that only short observation of single molecules remains possible.

For a given fluorophore density and diffusion constant, we investigated the effects of gate frequency and bleaching radius on the density of unbleached fluorophores inside the ROI. For instance, 0.5 Hz gating is sufficient to bring the density of fluorophores diffusing at $0.1\,\mu m^2\,s^{-1}$ below the detection limit, but $\sim 3$ Hz is required to achieve the same for faster-diffusing particles with $D = 1\,\mu m^2\,s^{-1}$ (Fig. 5e). When gating frequency is kept constant at 0.5 Hz, a smaller ROI leads to a larger effective fluorophore density in the ROI (Fig. 5f). Unlike TOCCSL, a typical PhotoGate experiment rapidly reaches a steady state, wherein the flux of unbleached molecules into the ROI is nearly balanced by the combined photobleaching by the TIRF beam and the gate beam (Fig. 5f; Supplementary Movies 10, 11, 12, and 13). As a result, PhotoGate keeps the density of fluorophores constant

throughout the experiment and can be used for a wide range of ROI sizes without sacrificing the SPT ability over minutes, while TOCCSL only provides a short usable window before the ROI is overrun by unbleached molecules. Our simulation software will allow users to optimize experimental parameters for a given biological system, resulting in high-throughput SPT at high fluorophore density.

## Discussion

PhotoGate introduces a concept of a gate beam that allows precise control over the number of fluorescent particles entering ROI and enables tracking of single particles at high spatial and temporal resolution until they photobleach or dissociate from their stationary target. This technical ability opens new possibilities for tracking single molecules in living cells at arbitrarily high concentrations without photoactivation. Using this method, we directly monitored transient binding of rapidly diffusing molecules to a stationary organelle. We also monitored diffusion and ligand-induced oligomerization of single protein complexes on a cell membrane at surface densities two orders of magnitude higher than the diffraction limit. The results demonstrate that PhotoGate is widely applicable to single-molecule dynamics of receptor signalling, membrane protein dynamics, endocytosis and cytoskeleton-associated proteins at high resolution. The method is also extensible to deep-cell imaging by scanning a non-diverging Bessel beam[27] in a circular pattern to gate the boundary of a cylindrical ROI.

PhotoGate offers several unique advantages over existing methods that achieve single-molecule tracking in dense speci-mens. Using TOCCSL, diffusion of single fluorescent clusters can be tracked on a cell membrane at the onset of the fluorescence recovery process[11]. This method enables multicolour tracking[28] and determines the diffusion constant[11] and brightness[29] of individual fluorophores. However, tracking time is limited by the surface density and diffusion constant of fluorescent particles in the unbleached region, and single particles typically cannot be resolved one second after photobleaching (Fig. 5)[11]. PhotoGate allows the tracking of individual particles in cellular processes that take longer than a few seconds (Fig. 5), such as arrival to and dissociation of APPL1 from an endosome (Fig. 2). Unlike TOCCSL, the tracking time of PhotoGate is insensitive to the diffusion constant and surface density of fluorescent particles, and is limited solely by the photon budget of the fluorophore (Fig. 5).

When diffusing proteins interact with a stationary structure and pause, such as in the case of APPL1 binding to endosomes, FRAP analysis is further complicated by the fact that multiple processes occurring at different rates (such as diffusion, binding and unbinding) contribute to the recovery signal. Fluorescence correlation spectroscopy has been used to specifically measure diffusion and filter out immobilized molecules[13]. PhotoGate is complementary to these methods, because it allows examination of bound molecules only, while the signal from the rapidly diffusing molecules is averaged out and thus excluded from analysis.

The major advantage of sptPALM over PhotoGate is the ability to obtain high-density single-molecule trajectories in a single cell by simultaneously activating, imaging and bleaching molecules across the cell[21]. In comparison, PhotoGate provides unique advantages by bypassing the requirement of using photo-convertible probes. First, our method provides higher spatial and temporal resolution, and longer tracking times (Figs 3 and 5), because the best-performing fluorescent proteins (such as eGFP and mNeon Green[30]) are superior to photoactivated fluorescent proteins in quantum yield, brightness and photostability[8]. We note that for applications demanding even higher brightness or

photostability, PhotoGate can be used to image organic dyes such as Cy3 or TMR without further modifications. Second, sptPALM, which relies on stochastic photoactivation of a small subset of fluorophores, is not suitable for counting the total number of fluorophores in a diffusing spot. In comparison, fluorophores observed via PhotoGate are in the bright state at the start of imaging, which allows the stoichiometry of individual spots to be determined by subunit counting (Fig. 4). Third, PhotoGate is more suitable for multicolour applications, because it uses the same wavelength for imaging and gating. In comparison, the probes that change their excitation and emission wave-length upon photoinduction (such as Kaede and Eos) require a second excitation beam at a different wavelength[31], often causing crosstalk between the fluorescent channels. Further-more, PhotoGate is applicable to existing cell lines that express conventional fluorescent probes and eliminates the requirement of constructing and testing new strains for single-molecule imaging.

In conclusion, PhotoGate is a superior method for long-term tracking, multicolour imaging and accurate detection of oligo-merization states of single molecules.

## Methods

**Sample preparation.** DNA encoding the full-length human EGFR gene with a C-terminal eGFP[A206K] fusion in pEGFP-N1 (Clontech) mammalian expression vector was stably transfected into COS-7 cells. Briefly, COS-7 cells were transiently transfected using FuGENE (Roche) according to the manufacturer's instructions. Stable clones were isolated by selection in 800 µg ml$^{-1}$ G418 for 4 weeks. Stable COS-7 cell lines were maintained in Dulbecco's Modified Eagle's Medium supplemented with 10% fetal bovine serum (FBS), streptomycin/penicillin and 200 µg ml$^{-1}$ G418. 48 h before use, cells were split into glass bottom petri dishes (Matek) with phenol-red free DMEM. 12 h before the experiment, cells were serum-starved by replacing the medium with FBS-free medium. When noted, cells were treated with or without 16 nM EGF for 5 min, before measurements were performed. The microscopy assays were performed at 37 °C.

U2OS cells were cultured following standard tissue culture protocols in McCoy's 5A media supplemented with 5% FBS. Twenty-four hours before imaging, 1.5 million U2OS cells were transfected via Nucleofector-T2 (Lonza) with 350 ng APPL1:GFP plasmid and split into two 35-mm glass-bottom poly-D-lysine-coated-imaging dishes (MatTek). Lonza Kit R nucleofection solution and pulse X-001 were used to transfect cells. Before imaging, cells were incubated for 40 min in imaging buffer supplemented with 5 mM glucose, amino acids and 5% dialyzed FBS, and containing 16.6 µM nocodazole to dampen endosomal trafficking via microtubules. Cells were imaged at 37 °C in the same buffer.

**Microscope.** An objective-type TIRF microscope was set-up, using a Nikon TiE-inverted microscope equipped with a perfect focusing unit, bright-field illumination and a × 100 1.49 numerical aperture PlanApo oil immersion objective (Nikon). A 488-nm solid state laser (coherent) was used for GFP excitation. The GFP signal was recorded by an Andor iXon 512 × 512 electron-multiplied charge-coupled device (EM-CCD) camera. Extra magnification ( × 1.5) was used to obtain a pixel size of 106 nm. Excitation laser beams were controlled by Uniblitz shutters. Because the CCD image is saturated under intense laser illumination, shutter timing was synchronized with the camera acquisition by a DAQ card (NI, USB-6221). In a continuous acquisition mode of the camera, frames that were exposed to a high-power PhotoGate beam was not used for data analysis. In a time-lapse data acquisition mode, the gate beam was turned on during the times when the camera was not acquiring data.

**The PhotoGate assay.** A focused 488 nm laser beam (2 MW cm$^{-2}$) was steered with a fast piezo-driven mirror (S-330.8SL, Physik Instrumente). The piezo-driven mirror was mounted at a position conjugate to the back-focal plane of the objective (Supplementary Fig. 1a) to ensure that the tilting of the mirror results in pure translation of the focused beam in the image plane. The mirror has an angular travel range of ~10 mrad (with a slight difference between the two axes) and provides a usable range of 22 µm by 30 µm at the image plane of the microscope. The mirror's angle was updated at 200 Hz via two analogue output channels of a USB-6221 DAQ card (National Instruments). The mirror was controlled by software custom-written in LabVIEW to define the dimension and the shape of the ROI. Typically, 50 outward spirals (each 0.2 s in duration) were used to pre-bleach the ROI, followed by a 2 s dark waiting period. The TIRF beam was then switched on and videos were recorded at rates between 0.67 and 20 Hz. The radius of the TIRF beam was dynamically adjusted using a variable-diameter iris at a point conjugate to the image plane (Supplementary Fig. 1a) to illuminate the ROI and

prevent bleaching in the rest of the cell. Imaging was interspersed by 200-ms long sweeps around the outer perimeter of the bleached area, repeated every 2 s. At acquisition rates slower than 5 frames per second, exposure was limited to 50 ms while the dark time between exposures was changed to result in the final desired frame rate.

**sptPALM assays.** Cells were split into glass bottom petri dishes (Mattek) and transfected with mEos2-EGFR using Lipofectamine 2000. Twenty-four hours post transfection the cells were serum-starved by replacing the medium with FBS-free medium for 12 h. Photo-switching of mEos2-EGFR was accomplished using 1.6 W cm$^{-2}$ 405 nm TIRF excitation beam for 200 ms. The activated molecules were imaged using 561 nm TIRF excitation beam at 100 ms per frame. The power of the excitation beam was adjusted to 160 W cm$^{-2}$ to collect photons counts per spot similar to that of PhotoGate assays.

**Data analysis.** The probability of particles escaping the gate was calculated based on the time dependent recovery profile in FRAP experiments under the assumptions that all of the molecules undergo pure lateral diffusion with the same diffusion constant and that the gate beam is perfectly collimated and bleaches all of the particles located in the ring. The few molecules that were not bleached during pre-bleaching of the ROI were not included in data analysis.

The position of fluorescent spots was determined by fitting the PSFs to the 2D Gaussian function. The positions were fitted throughout the movie except at the frames when the photobleaching events happened, or the frames in which the tracked particle overlapped with other fluorophores. The intensity of the spots is estimated by the volume of the 2D Gaussian peak. In a typical assay, we adjusted excitation power to achieve 20-nm localization accuracy at 10 Hz image acquisition rate. Under these conditions, individual mNeonGreen molecules were tracked for 5 s on average.

2D diffusion of EGFR spots were analysed by MSD analysis. MSD plots of diffusing particles were fitted to a polynomial function, $\langle x^2 \rangle = 4Dt^a$, where $D$ is the diffusion constant. The first ($\alpha = 1$) and second ($\alpha = 2$) order polynomial fits represent Brownian motion and unidirectional transport, respectively (Supplementary Fig. 6). $\alpha < 1$ was interpreted as sub-diffusion due to the confinement of molecules. Ninety-three percentage of the trajectories fit well ($R^2 > 0.95$) to a linear function with a positive $y$ intercept. Other trajectories were excluded from data analysis to filter out the non-diffusive particles in our data set. Histograms are fitted by maximum likelihood estimator using original datasets. Pausing in EGFR diffusion is defined as the duration at which the s.d. of the position is $< 50$ nm within $> 1$ s.

APPL1 residence times were analysed by plotting the intensity versus time profile for each fluorescent spot that appeared in the ROI while PhotoGate was on. Background intensity traces were collected for each fluorescent spot from a nearby dark region and subtracted from the APPL1 traces to correct for photobleaching of the sample under TIR illumination. Some endosomes were not perfectly immobilized, and any spots that moved by more than five pixels before bleaching were excluded from further analysis. Each remaining intensity profile was inspected, with fluorophore arrival and departure times assigned manually. Traces, in which the arrival and departure times could not be identified due to background fluctuations, were excluded from the analysis. The cumulative probability distribution of dwell times was then fit to a single exponential model, yielding the characteristic off-rate.

For FRAP analysis, a circular ROI (1.5 μm in diameter) was manually drawn over a recovering endosome and an identically sized ROI placed in a bleached area that did not contain any endosomes. Fluorescence intensity trajectories were then obtained for each of the two spots and fitted with a single exponential recovery model. Cells without a large enough endosome-free region throughout the timeline of recovery were excluded from further analysis.

**Numerical simulations.** The computer simulation of the PhotoGate and TOCCSL experiments was written as a single module in Python 3.4 and is made freely available at http://research.physics.berkeley.edu/yildiz/SubPages/code_repository.html. Diffusion of particles on the membrane of a cell is modelled as a 2D random walk on a square lattice with a small time step, 0.004 ~ 0.016 s in duration[32]. The cell is defined as a 40 by 40 μm square, which at the start of the simulation is randomly and uniformly populated by unbleached fluorophores at the desired density (50 fluorophores per μm$^2$ in all our simulations). With each step of the simulation, every particle takes a step of size $(4D\tau)^{1/2}$, $D$ being the diffusion constant and $\tau$ the time step of the simulation. The step is taken in one of four directions (up, down, left or right) based on a uniform random number. If a particle attempts to leave the boundaries of a cell, it is forced to take a backwards step instead.

Illumination intensities are determined separately for TIRF illumination, which is modelled as a circular region of uniform intensity in the centre of the modelled cell and of the same radius as the ROI, and the gate beam itself. Similar to the experimental measurements, the region outside the ROI is not illuminated in PhotoGate simulations. To calculate intensities of the circular gate beam, we

numerically integrate the intensity of a Gaussian spot swept around in a full circle:

$$I(d) = \int_0^{2\pi} a e^{-(R^2 + d^2 - 2Rd\cos\theta)/2c^2} \, d\theta,$$

where $I$ is the intensity at the point of interest, $\theta$ is the angular position along the ring, $R$ is the radius of the ring, $d$ is the distance of the point of interest from the centre of the ring, $a$ is the amplitude of the Gaussian beam and $c$ is the width of the Gaussian beam. We used beam parameters typical in our optical set-up (Supplementary Tables 1–3 for exact parameter values). All TIRF intensities are represented in average number of photons emitted from a fluorophore per second.

Photobleaching as a function of illumination intensity is calculated at each step of the simulation for each fluorophore with a Monte Carlo algorithm. The probability of bleaching, on the interval from zero to one, is computed for each particle as determined by the local fluorescence intensity at the particle's position. Then, a uniform random number from zero to one is generated and compared with the pre-calculated probability of bleaching. If the random number is greater than the probability, the particle bleaches. Random numbers are drawn independently for each fluorophore in a particle in cases when particles are modelled to be oligomeric.

To investigate the suitability of PhotoGate for long-term tracking applications, we defined what constitutes a trackable trajectory. Any unbleached fluorescent particle was considered trackable in a given frame if it remained farther than 0.5 μm (twice the diffraction limit) apart from all other unbleached particles. Every sequence of consecutive frames over which this condition was satisfied was considered to be a single trackable trajectory. Trajectories that end by permanent photobleaching is used for subunit counting. Trajectories also end if the particle approaches another fluorescent particle too closely or exit the ROI before photobleaching, making them indistinguishable. These particles are not used in subunit counting.

**Data availability.** All raw data that support the findings of this study are available from the corresponding author upon reasonable request.

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

## Acknowledgements

We would like to thank J. Kuriyan and F.B. Cleary for helpful discussion and revision of our manuscript. This work has been supported by NIH (GM094522 (A.Y.)), the NSF Career Award (MCB-1055017 (A.Y.)), an NSF Graduate Research Fellowship (DGE 1106400 (V.B.)) and an HHMI International Student Research Fellowship (Y.H.).

## Author contributions

S.-M.S., V.B. and A.Y. designed experiments, Y.H., R.E.L. and J.B. performed cloning and mammalian cell cultures, S.-M.S. and V.B. built the microscope and performed single-molecule fluorescence assays, J.B. performed the sptPALM assays, and A.Y., S.-M.S., V.B. and R.Z. wrote the manuscript.

## Additional information

**Competing financial interests:** The authors declare no competing financial interests.

