## [Peer Review File · Nature Communications]

Review: **“PhotoGate Microscopy to Track Single Molecules in Crowded Environments”** by Vladislav Belyy, Sheng-Min Shih, Jigar Bandaria, Yongjian Huang, Rosalie E. Lawrence, Roberto Zoncu and Ahmet Yildiz.

In this manuscript the authors describe a novel approach to the dynamic measurement of single molecules in cells. They combine total internal reflection fluorescence microscopy measurements with fluorescence photobleaching to thin out the concentration of molecules in cells. This is often required to allow for convenient measurement of single molecule fluorescence from recombinantly expressed fluorescent protein constructs in cells. However, traditionally, this approach of bleaching and subsequent single molecule measurements usually allows for tracking on molecules for few seconds only before they bleach and new molecules can enter the region of interest (ROI) only via the border of the area and are continuously bleached as well. What is new in the approach presented by the authors is that they allow for reentry of molecules while not imaging, but then rebleach a ring-shaped area around the ROI to limit the amount of molecules in the ROI to an amount compatible with single molecule measurements that is at the same time distributed evenly over the ROI.

This is excellent work by a world-class single molecule microscopy group and the assay they present indeed to my knowledge allows for the first time to measure single molecule kinetics in cells. Also, it makes single molecule tracking easier. Positive is also that there are commercial systems that allow the realization of such assays. A drawback is that specialized equipment is required. The manuscript is well-written and the data are convincing. It is absolutely suitable for Nature communications and should be published.

Minor comments:

- 1) Is the scale bar in Figure 3e & g correct? That area must be a very large fraction of the ventral surface of the cell.

Response to Reviewers:

Reviewer 4

Is the scale bar in Figure 3e & g correct? That area must be a very large fraction of the ventral surface of the cell.

The scale bar is correct. In panel e, the ROI occupies approximately a fifth of the cell's surface. As we demonstrate in the manuscript, the diameter of the ROI does not significantly affect our ability to detect and track single molecules, and we simply chose a large ROI in a large cell for illustration purposes. The images in panel g, on the other hand, come from an sptPALM experiment, where the fluorescent proteins are stochastically activated throughout the cell and are not constrained to an ROI.